# Governmental Anti-Pandemic and Subsidy Strategies for Blockchain-Enabled Food Supply Chains in the Post-Pandemic Era

Changhua Liao [ID], Qihui Lu *[ID] and Ying Shui

School of Business Administration, Zhejiang Gongshang University, Hangzhou 310018, China; changhual@163.com (C.L.); shui_ying_0730@163.com (Y.S.)
* Correspondence: qihuilu@zjgsu.edu.cn

**Abstract:** Aiming to explore whether governments should choose anti-pandemic or subsidy strategies in the post-pandemic era, this study constructed a three-level food supply chain that was composed of a leading third-party logistics provider, a supplier and a retailer, in which the third-party logistics provider used blockchain technology for food traceability to address consumer concerns about food safety. We then used game theory to analyze the pricing decisions, traceability levels, anti-pandemic effort levels and subsidy levels of the supply chain under different governmental anti-pandemic or subsidy strategies. Our results showed that in all scenarios, the higher the consumer preference for traceability information, the larger the traceability levels and anti-pandemic effort levels and the more favorable the outcome for all parties; thus, governments should improve consumer awareness of pandemic prevention. For the benefit of all parties, governments should adopt anti-pandemic and subsidy strategies simultaneously, even in the post-pandemic era. Interestingly, for the scenario in which governments could only adopt one strategy, when the cost coefficient of traceability was small, the governmental subsidies would actually lead to lower traceability levels of the 3PL. This study could provide decision-making references for governments during the post-pandemic era and a new possibility for blockchain application.

**Keywords:** three-level food supply chain; blockchain technology; governmental strategy selection; traceability; game theory



## 1. Introduction

In reality, the worldwide consumption of fresh products, including seafood, eggs and meat, is enormous. With the development of cold chain technologies, enterprises usually use cold chain logistics to transport these products, so that consumer demand for fresh products can be satisfied. Due to the high specialization and standardization of third-party cold chain logistics, the wide coverage of logistics networks and the low associated costs, most fresh produce companies tend to outsource their logistics operations to third-party cold chain logistics companies [1]. While cold chain logistics has greatly increased the quality of fresh produce, food safety has been challenged by the COVID-19 pandemic. Consumers now have serious health and safety concerns about the origin and transportation of their food, within which viruses can survive for long periods of time [2]. For example, in Alibaba-owned Hema Fresh supermarkets, several workers were infected while processing food that was supplied via cold chain logisticsand so, the imported frozen meat and seafood products that were sold through Hema Fresh had to be subjected to strict testing. Therefore, in order to rebuild consumer trust during the post-COVID-19 pandemic era, supply chain traceability and transparency are critical to the survival of businesses [3].

Blockchain technology is an emerging traceability technology that is decentralized, open and transparent, cryptographically protected and tamper-proof [4]. Compared to

other traceability technologies, such as RFID technology, blockchain technology has unparalleled advantages for product traceability. Supply chain node companies are able to not only use blockchain technology to monitor product quality in real time, but also effectively track and identify potentially infected products [5]. Over recent years, some industry giants have pioneered the use of blockchain technology to track product information, including manufacturing companies, logistics companies and e-commerce companies. For example, Maersk has used blockchain technology to help shippers, ports, customs, banks and logistics providers to track their shipments. Alibaba has subsequently launched the Tmall e-commerce platform, which uses blockchain technology and is based on a cold chain traceability system [1,6].

By leveraging the advantages of blockchain, enterprises can increase the purchasing power of consumers so that the enterprises can weather this storm on their own. For governments, appropriate strategies should also be adopted to address consumer concerns and improve supply chain performance. Since the end of 2019, COVID-19 has ravaged the world, with food supply chains being severely affected. When the pandemic was at its most severe, many countries throughout the world formulated emergency rescue measures. Cities that were hit hard by the outbreak enforced "city closures" to deal with the pandemic and many internet companies allowed employees to work from home [7]. However, in the post-pandemic era, the pandemic has eased but not completely disappeared. The rates of infection now rise and fall from time to time and small-scale outbreaks can occur at any time. During this era, many countries throughout the world have begun to cancel or reduce the levels of their pandemic prevention measures [8]. After experiencing the pandemic, most people's concept of life has changed and they are now more concerned about food safety and the social consumption capacity has declined. In order to improve consumer purchasing levels and avoid socioeconomic imbalances, governments could either continue to adopt anti-pandemic strategies or subsidize blockchain traceability technologies to address people's concerns about food safety, thereby improving social production levels and the economic environment. Government subsidies come in a variety of forms; for example, the Government of India's Ministry of Power subsidizes the purchase of LED light bulbs across the country [9]. In this study, we focused on subsidies for blockchain traceability, which could also be regarded as a kind of governmental subsidy for enterprises to address consumer concerns about food safety and could help governments to fight the ongoing effects of the pandemic.

Within the above context, we know that blockchain traceability technologies and governmental anti-pandemic policies are very important to supply chain operations during the post-pandemic era, but the costs of anti-pandemic and traceability strategies are high. At this point, whether governments should choose to continue their own investments into anti-pandemic efforts, provide subsidies to allow enterprises to improve traceability or choose both anti-pandemic and subsidy strategies simultaneously is worthy of attention. Therefore, we investigated the following important research questions:

(1) When adopting blockchain technology, what are the optimal decisions for food supply chains both with and without governmental anti-pandemic or subsidy strategies?

(2) In the post-pandemic era, should governments continue to adopt anti-pandemic strategies or new subsidy strategies? Which strategies would be better?

In order to solve the above problems, we first studied four scenarios in which governments adopted anti-pandemic strategies or subsidy strategies. We found that anti-pandemic strategies were only established when the cost coefficients of the anti-pandemic efforts were too large and that subsidy strategies were only established when the cost coefficients of traceability were too large. Then, we analyzed the impacts of consumer preference for traceability information on equilibrium solutions, the profits of supply chain members, consumer surplus and social welfare. The results showed that in all scenarios, the greater the consumer preference for traceability information, the more beneficial the outcomes for all parties and society. Finally, by comparing the various scenarios, we found that the simultaneous adoption of anti-pandemic and subsidy strategies by governments would

always be beneficial to suppliers, retailers and consumers and that the impacts on third-party logistics providers (3PL) and social welfare would be related to the cost coefficients of anti-pandemic efforts and the consumer preference for traceability information. In other words, it would be the most detrimental to supply chain members and society in general for governments not to adopt any strategies.

To the best of our knowledge, by considering the consumer preference for traceability information, this is the first work to investigate governmental anti-pandemic and subsidy strategies in blockchain-enabled food supply chains (FSCs). Specifically, this study focused on answering the question of whether governments should choose anti-pandemic strategies or subsidy strategies when considering the costs of anti-pandemic efforts and subsidies for blockchain traceability technologies. Our research uncovered the effects of governmental strategy selection on traceability information levels based on blockchain technology, food prices in the post-pandemic era, the profits of the supply chain members, consumer surplus and social welfare. These could provide managerial implications for the operational decisions of blockchain-enabled FSCs and provide decision-making references for governmental anti-pandemic efforts when adopting blockchain technology.

The remainder of this paper is organised as follows. Section 2 presents a review of the related literature. Section 3 provides the model description. Section 4 constructs the model and then derives the equilibrium solutions. Section 5 provides a comparison among the different models. Section 6 states practical and theoretical implications. Section 7 summarizes our conclusions. All the proofs are provided in the Appendix A.

## 2. Literature Review

This study is related to three streams of literature, namely, the application of blockchain technology in supply chain management, food traceability, and the governmental anti-pandemic and subsidy strategies.

### 2.1. Application of Blockchain Technology in Supply Chain Management

In recent years, the application of blockchain technology in supply chain management has been studied by many scholars (Fan et al. [10]; Chang et al. [11]; Liu et al. [12]; Wong et al. [13]; Dubey et al. [14]; Pun et al. [15]; Niu et al. [16]; Wang et al. [17]; Xu et al. [18]). For example, Choi [19] analyzed different consumer utility driven operations models and highlighted the values of blockchain technology supported platforms for diamond authentication and certification. Choi and Luo [20] studied the blockchain adoption to help and identify the situation in which blockchain helps enhance social welfare but brings harm to supply chain profitability. Choi et al. [21] explored the value of blockchain technology mediated customized service pricing strategy. De Giovanni [22] built a supply chain composed of a supplier and a retailer, highlighting where the use of smart contracts made blockchain applications more operationally convenient and economically appealing. Under cap-and-trade regulation, Xu and Choi [23] proposed a supply chain composed of an offline manufacturer and an online platform. They found that whether blockchain technology could increase profitability of the manufacturer's offline channel depended on the cross-channel effect. Niu et al. [24] studied a two-period model where two private hospitals compete in both service quality and service charges. They suggested that to save patients' costs, blockchain-based health information exchange for differentiated service should be adopted. Tao et al. [25] analyzed the effect of blockchain on optimal pricing and quality decisions under two different supply chain structures. They found that when consumer acceptance of blockchain is low, the products with blockchain have not only a low price but also a high quality in the single platform. Wu and Yu [26] investigated the impact of blockchain technology on platform supply chains from the perspectives of information transparency and transaction cost. Their results indicated that the blockchain can be a great tool for platforms to control the market and adjust prices. In view of the food supply chains with health-safety concerns in the post-pandemic era, considering the traceability and transparency of reliable information supported by blockchain implementation, we explore

the impact of blockchain on operational decisions such as FSCs pricing and traceability levels, which enriches the existing research.

### 2.2. Food Traceability

Nowadays, consumers are paying more attention to food safety which has aroused the attention of scholars. For example, Shankar et al. [27] analyzed the effect of consumer satisfaction, effective traffic management, manufacturer brand and government regulations, among other factors, on the traceability of the food logistics system. Haleem et al. [28] explored the factors influencing the implementation of traceability systems in food supply chains. They found that food safety and quality are the most important factors. Similarly, Casino et al. [29] showed that the blockchain and IoT is beneficial in building a secure food traceability system. In addition, some scholars studied the application of blockchain technology to food traceability. Tan et al. [30] proposed a blockchain-based traceability framework that had been tested and verified in the real-life halal food supply chain. Rogerson and Parry [31] believed that blockchain adoption in a food supply chain can increase consumer trust and heighten visibility. Qian et al. [32] and Stranieri et al. [33] dissected the application, advantages, and future challenges of blockchain in food traceability. It should be point out that, most of the above research analyzed the issue of food traceability from a theoretical exposition or empirical research perspective. Liu et al. [34] was the first to use a game-theoretic model to show the positive impact of blockchain on fresh food traceability and the impact of food marketing effect generated by the blockchain traceability. Niu et al. [35] further provided the value of blockchain in identifying the responsible party for bacteria pollution. Moreover, Yang et al. [6] focused on the food supply chains with health-safety concerns, and applied the traceability and transparency of reliable information supported by blockchain. Wu et al. [1] studied the optimal strategies of the three-level fresh product supply chain when considering the blockchain-based traceability system. Similar to Yang et al. [6] and Wu et al. [1], we study a blockchain-based food traceability system dominated by the 3PL, taking into account consumers' concerns about epidemic infections and their preferences for traceability information. The difference with our research is that we focus on the governmental strategic choices in the post-pandemic era.

### 2.3. Governmental Anti-Pandemic and Subsidy Strategies

This paper also relates to the governmental anti-pandemic and subsidy strategies. For the study of epidemic prevention, Chinazzi et al. [36] projected the effect of travel limitations on the national and international spread of the epidemic by using a global metapopulation disease transmission model. Liao and Wang [37] studied the effect of epidemic information and rumors on public's worries and attitude toward prevention measures during the outbreak of COVID-19. Their results showed the importance of timely and credible information providing to evoke a certain level of worry and promote public cooperation. Wu et al. [38] indicated how China responded to the public health emergency of COVID-19 from the perspective of policy making. They showed that the Chinese government adopted a multi-agency, joint epidemic prevention and control mechanism to ensure the efficiency of policy implementation. It can be found that the research on epidemic prevention is mainly biased towards qualitative analysis. Research on governmental subsidy is common in low-carbon supply chains and green supply chains (Wang et al. [39]; Li et al. [40]; Zhang et al. [41]). Xue et al. [42] used a game-theoretical model to analyze the green product design and multi-product pricing strategies under different government subsidy strategies and supply chain structures. They stated that to realize the sustainable development of our society, the enterprise should adopt reasonable measures to boost the environmental awareness of consumers. Ma et al. [43] explored the effect of government subsidy on participants' performances under asymmetric carbon reduction information. They found that if subsidy policy is more effective in stimulating demand, the subsidy can encourage the manufacturer to share information with the government. Xu and Duan [44] studied optimal strategies for pricing and greenness investment for green products with

government subsidies and probed the conditions for adopting blockchain. In addition, Yang and Qian [7] examined the equilibrium results and anti-epidemic effort level of supply chain under different government subsidy measures and coordination strategies. They found that in the context of the epidemic, government subsidies can improve the level of social welfare. We also consider the governmental subsidy strategy, but it is a subsidy for the 3PL using blockchain traceability. At the same time, we consider the government to decide the anti-pandemic effort level, and explore what strategies the government should adopt to deal with the epidemic in the post-pandemic era.

### 2.4. Research Gap

In this subsection, we summarize the research gaps between our work and the existing literature and further highlight our contributions. First, to the best of our knowledge, this paper is the first to use the game theory model to describe the importance of blockchain technology to food traceability in the context of the COVID-19. We explore the government's choice of two strategies for anti-pandemic and subsidy under the blockchain traceability. Second, this paper considers the 3PL as the leader of the supply chain, which bears the cost of blockchain traceability and determines the traceability level, and explores the impact of different governmental strategies on the traceability levels and profits. Third, considering the government's decision to maximize social welfare, we study the impact of different scenarios on society and consumers, so as to provide decision-making references for the government in the post-pandemic era and a new background for blockchain application. This is not only conducive to the stability and sustainable development of the supply chain, but also to the mitigation of the epidemic.

### 3. Model Description

We consider a food supply chain system consisting of one supplier, one 3PL and one retailer, who are indexed by *S*, *L* and *R*, respectively. The supplier outsources the logistics service to the 3PL at a unit logistics service price *m* and sells this product to the retailer via a wholesale contract at a unit wholesale price *w*. The 3PL delivers the products to the retailer, and then the retailer sells the products to consumers at a unit retail price *p*, with $p > w > m > 0$. To simplify the calculation, we assume that the supplier's production cost and the 3PL's logistics service cost are zero [16,45]. For a consumer in the market, the valuation towards the product under normal circumstances is denoted by *v*, which is a continuous variable and follows a density function $f(v)$. We assume that *v* follows a uniform distribution in the range of 0 and 1 [1,6,10].

In the post-pandemic era, the epidemic occurs occasionally, thus the supply chain faces an infection risk. For example, the areas through which the transport passes may be experiencing a COVID-19 outbreak, which exposes the product to an infection risk during the logistics process. Facing the infection risk, consumers become concerned about the safety of the product. To address these concerns and meet consumers' need for authentic and reliable product traceability information, one efficient approach is adopting blockchain given its information transparency and traceability. Under the scenario where the FSCs adopt blockchain, more information will lead to higher degree of trust for consumers who have preference for traceability information and higher consumers' utility, but more traceability information means higher traceability cost [1]. We denote the traceability information level based on blockchain as *t*. Thus, following [1,6,46], we assume that the cost of traceability by using blockchain is $\frac{1}{2}kt^2$, and the consumers' utility function is $u = v - bp + \delta t$, where $k > 0$ indicates the cost coefficient of traceability by using blockchain, $b > 0$ is the price sensitivity of the product. The implementation of blockchain does not directly influence the product characteristics and price sensitivity but positively affects the utility by fostering consumers' trust. $\delta t$ represents the increase of traceability level on consumers' utility, $\delta > 0$ is the consumer preference for traceability information.

The consumers will purchase the product only if $v - bp + \delta t > 0$. Thus, the demand function under the scenario of adopting blockchain can be formulated as follows:

$$D^B = \int_{bp-\delta t}^{1} f(v)dv = 1 - bp + \delta t. \tag{1}$$

In the presence of the pandemic, for the health and safety of the people, the government will make efforts to prevent the epidemic, at this time, we denote the anti-pandemic effort level for the government as $e^p$. In the post-pandemic era, the epidemic is not as severe as in the previous stage, so that the government has successively loosened the requirements for anti-pandemic and lowered the anti-pandemic effort level, we denote the decrease of anti-pandemic effort level as $e^d$. Thus, in the post-pandemic era, the governmental actual anti-pandemic effort level is $e = e^p - e^d$, where $e = 0$ indicates the government will not adopt anti-pandemic strategy. A high level of anti-pandemic effort means high anti-pandemic effort costs. Thus, if the government adopts the anti-pandemic effort level $e$, it involves a cost $\frac{1}{2}he^2$, where $h$ is the cost coefficient of the anti-pandemic effort. Such a quadratic-type function is commonly used in previous studies [7,47,48]. In addition, the higher the anti-pandemic effort level of the government, the lower the probability of products being infected, thereby reducing consumers' concerns about products and improving consumers' perceived utility of products. Therefore, when the government adopts the anti-pandemic strategy, consumers' utility function can be described as $u = v - bp + \theta e + \delta t$ [7], where $\theta e$ represents the increase of anti-pandemic effort level on consumers' utility, and without loss of generality, we set $\theta = 1$ [6]. The consumers will purchase the product only if $v - bp + e + \delta t > 0$. Thus, the demand function under the scenario of adopting blockchain and anti-pandemic strategy can be formulated as follows:

$$D^E = \int_{bp-e-\delta t}^{1} f(v)dv = 1 - bp + e + \delta t. \tag{2}$$

In addition, in the post-pandemic era, the government can adopt the subsidy strategy in addition to the anti-pandemic strategy. That is, the government can subsidize the blockchain traceability level per unit to encourage supply chain members to improve the blockchain traceability level, ultimately solving consumers' concerns about product safety and improving consumers' utility. We denote the subsidy amount as $st$, where $s > 0$ represents the unit subsidy level. This governmental subsidy strategy is common in low-carbon supply chains and green supply chains [39,40]. As for whether the government adopts an anti-pandemic strategy or a subsidy strategy, it can be divided into four scenarios according to the benefits and costs brought by different strategies to the society and the supply chain. We use subscript $j \in \{1, 2, 3, 4\}$ to denote different scenarios. Specifically, $j = 1$ denotes the scenario where the government does not adopt anti-pandemic and subsidy strategies, i.e., Model-1; $j = 2$ denotes the scenario where the government adopts anti-pandemic strategy and does not adopt subsidy strategy, i.e., Model-2; $j = 3$ denotes the scenario where the government adopts subsidy strategy and does not adopt anti-pandemic strategy, i.e., Model-3; $j = 4$ denotes the scenario where the government adopts anti-pandemic and subsidy strategies, i.e., Model-4. The key parameters and decision variables are summarized in Nomenclature.

In the multi-level food supply chain, the supplier, 3PL and retailer are in a Stackelberg game, where the 3PL, supplier and retailer are the leader, sub-leader and follower, respectively. As the leader, the 3PL will bear the blockchain traceability cost and determine the traceability level. Therefore, governmental subsidy only subsidizes the 3PL. The decision sequence under the scenario where the 3PL adopts blockchain technology and the government simultaneously adopts anti-pandemic and subsidy strategies is shown in Figure 1. First, the government simultaneously decides the anti-pandemic effort level $e$ and subsidy level $s$. Then, the 3PL simultaneously determines the traceability level $t$ and logistics service price $m$. Finally, the supplier determines the wholesale price $w$, and the

retailer determines the retail price $p$. In addition, the supply chain members make decisions with the goal of maximizing their own profits, while the government makes decisions with the goal of maximizing social welfare.

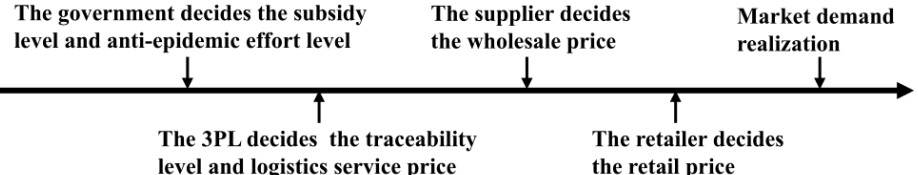

**Figure 1.** The event sequence.

## 4. Model Analysis

In this section, we first analyse the optimal pricing strategy of FSC members, the optimal traceability level of the 3PL, and the optimal anti-pandemic effort level and subsidy level of the government under the four scenarios. Then, we explore the impact of consumer preference for traceability information on the profits and decisions of all parties. Finally, the profits of supply chain members are compared.

### 4.1. Model-1

In this case, the government neither invests in anti-pandemic efforts nor implements subsidy strategy in the post-pandemic era. The sequence of events is illustrated in Figure 1. The 3PL, supplier, and retailer's profits are, respectively,

$$\Pi_{L1} = m_1(1 - bp_1 + \delta t_1) - \frac{1}{2}kt_1^2, \tag{3}$$

$$\Pi_{S1} = (w_1 - m_1)(1 - bp_1 + \delta t_1), \tag{4}$$

$$\Pi_{R1} = (p_1 - w_1)(1 - bp_1 + \delta t_1). \tag{5}$$

In addition, consumer surplus and social welfare can be formulated as, respectively,

$$CS_1 = \int_{bp_1 - \delta t_1}^{1} (v - bp_1 + \delta t_1)f(v)dv, \tag{6}$$

$$SW_1 = \Pi_{L1} + \Pi_{S1} + \Pi_{R1} + CS_1. \tag{7}$$

Using the backward method to solve Equations (3)–(5) in turn, we can obtain the following lemma.

**Lemma 1.** *Under Model-1, if $k > \frac{\delta^2}{8b}$, the optimal strategies of supply chain are $w_1^* = \frac{6k}{8bk - \delta^2}$, $m_1^* = \frac{4k}{8bk - \delta^2}$, $p_1^* = \frac{7k}{8bk - \delta^2}$, $t_1^* = \frac{\delta}{8bk - \delta^2}$, respectively.*

Lemma 1 shows that each member of the supply chain (i.e., the supplier, 3PL and retailer) has a unique optimal strategy only if $k > \frac{\delta^2}{8b}$. The condition ($k > \frac{\delta^2}{8b}$) means that under Model-1, the cost of blockchain traceability must be relatively large. When this condition fails to hold, there can be multiple groups of strategies. For the sake of simplicity, we only focus on the unique optimal strategy.

According to Lemma 1, the demand and optimal profits of the supplier, 3PL and retailer under the Model-1 are as follows:

$$D_1^{B*} = \frac{bk}{8bk - \delta^2},$$

$$\Pi_{S1}^* = \frac{2bk^2}{(8bk - \delta^2)^2},$$

$$\Pi_{L1}^* = \frac{k}{2(8bk - \delta^2)},$$

$$\Pi_{R1}^* = \frac{bk^2}{(8bk - \delta^2)^2}.$$

Furthermore, we can derive the consumer surplus and social welfare are, respectively,

$$CS_1^* = \frac{b^2 k^2}{2(8bk - \delta^2)^2},$$

$$SW_1^* = \frac{k(14bk - \delta^2 + b^2 k)}{2(8bk - \delta^2)^2}.$$

*4.2. Model-2*

In this case, the government adopts anti-pandemic strategy, but does not adopt subsidy strategy in the post-pandemic era. The sequence of events is illustrated in Figure 1. The 3PL, supplier, and retailer's profits are, respectively,

$$\Pi_{L2} = m_2(1 - bp_2 + e_2 + \delta t_2) - \frac{1}{2}kt_2^2, \tag{8}$$

$$\Pi_{S2} = (w_2 - m_2)(1 - bp_2 + e_2 + \delta t_2), \tag{9}$$

$$\Pi_{R2} = (p_2 - w_2)(1 - bp_2 + e_2 + \delta t_2). \tag{10}$$

In addition, the consumer surplus and social welfare can be formulated as, respectively,

$$CS_2 = \int_{bp_2 - e_2 - \delta t_2}^{1} (v - bp_2 + e_2 + \delta t_2) f(v) dv, \tag{11}$$

$$SW_2 = \Pi_{L2} + \Pi_{S2} + \Pi_{R2} + CS_2 - \frac{1}{2}he_2^2. \tag{12}$$

Using the backward method to solve Equations (8)–(10) and (12) in turn, we can obtain the following lemma.

**Lemma 2.** *Under Model-2, if* $k > \frac{\delta^2}{8b}$ *and* $h > \frac{k(14bk - \delta^2 + b^2 k)}{(8bk - \delta^2)^2}$, *the optimal strategies of the multi-level supply chain are* $e_2^* = \frac{k(14bk - \delta^2 + b^2 k)}{h(8bk - \delta^2)^2 - k(14bk - \delta^2 + b^2 k)}$, $w_2^* = \frac{6kh(8bk - \delta^2)}{h(8bk - \delta^2)^2 - k(14bk - \delta^2 + b^2 k)}$, $m_2^* = \frac{4kh(8bk - \delta^2)}{h(8bk - \delta^2)^2 - k(14bk - \delta^2 + b^2 k)}$, $p_2^* = \frac{7kh(8bk - \delta^2)}{h(8bk - \delta^2)^2 - k(14bk - \delta^2 + b^2 k)}$, $t_2^* = \frac{h\delta(8bk - \delta^2)}{h(8bk - \delta^2)^2 - k(14bk - \delta^2 + b^2 k)}$, *respectively.*

Lemma 2 shows that each member of the supply chain and the government have a unique optimal strategy only if $k > \frac{\delta^2}{8b}$ and $h > \frac{k(14bk - \delta^2 + b^2 k)}{(8bk - \delta^2)^2}$. The condition $(k > \frac{\delta^2}{8b})$ means that under Model-2, the cost of blockchain traceability must be relatively large and be the same as Model-1. The condition $(h > \frac{k(14bk - \delta^2 + b^2 k)}{(8bk - \delta^2)^2})$ means that under Model-2, the cost of the governmental anti-pandemic efforts must be relatively large. If $h$ is too small, there is no need to explore the optimality and the government will always increase investment in anti-pandemic efforts.

According to Lemma 2, we can determine the demand and optimal profits of the supplier, 3PL and retailer under the Model-2 are as follows:

$$D_2^{E*} = \frac{bkh(8bk - \delta^2)}{h(8bk - \delta^2)^2 - k(14bk - \delta^2 + b^2k)},$$

$$\Pi_{S2}^* = \frac{2bk^2h^2(8bk - \delta^2)^2}{(h(8bk - \delta^2)^2 - k(14bk - \delta^2 + b^2k))^2},$$

$$\Pi_{L2}^* = \frac{kh^2(8bk - \delta^2)^3}{2(h(8bk - \delta^2)^2 - k(14bk - \delta^2 + b^2k))^2},$$

$$\Pi_{R2}^* = \frac{bk^2h^2(8bk - \delta^2)^2}{(h(8bk - \delta^2)^2 - k(14bk - \delta^2 + b^2k))^2}.$$

Furthermore, we can derive the consumer surplus and social welfare are, respectively,

$$CS_2^* = \frac{b^2k^2h^2(8bk - \delta^2)^2}{2(h(8bk - \delta^2)^2 - k(14bk - \delta^2 + b^2k))^2},$$

$$SW_2^* = \frac{kh(14bk - \delta^2 + b^2k)}{2(h(8bk - \delta^2)^2 - k(14bk - \delta^2 + b^2k))}.$$

*4.3. Model-3*

In this case, the government adopts subsidy strategy, but does not adopt anti-pandemic strategy in the post-pandemic era. The sequence of events is illustrated in Figure 1. The 3PL, supplier, and retailer's profits are, respectively,

$$\Pi_{L3} = m_3(1 - bp_3 + \delta t_3) - \frac{1}{2}kt_3^2 + s_3t_3, \tag{13}$$

$$\Pi_{S3} = (w_3 - m_3)(1 - bp_3 + \delta t_3), \tag{14}$$

$$\Pi_{R3} = (p_3 - w_3)(1 - bp_3 + \delta t_3). \tag{15}$$

In addition, the consumer surplus and social welfare can be formulated as, respectively,

$$CS_3 = \int_{bp_3 - \delta t_3}^1 (v - bp_3 + \delta t_3)f(v)dv, \tag{16}$$

$$SW_3 = \Pi_{L3} + \Pi_{S3} + \Pi_{R3} + CS_3 - s_3t_3. \tag{17}$$

Using the backward method to solve Equations (13)–(15) and (17) in turn, we can obtain the following lemma.

**Lemma 3.** *Under Model-3, if* $k > \frac{(14+b)\delta^2}{64b}$, *the optimal strategies of supply chain are* $s_3^* = \frac{k\delta(6+b)}{64bk - \delta^2(14+b)}$, $w_3^* = \frac{48k}{64bk - \delta^2(14+b)}$, $m_3^* = \frac{32k}{64bk - \delta^2(14+b)}$, $p_3^* = \frac{56k}{64bk - \delta^2(14+b)}$, $t_3^* = \frac{\delta(14+b)}{64bk - \delta^2(14+b)}$, *respectively.*

Lemma 3 shows that each member of the supply chain and the government have a unique optimal strategy only if $k > \frac{(14+b)\delta^2}{64b}$. The condition ($k > \frac{(14+b)\delta^2}{64b}$) means that under Model-3, the cost of blockchain traceability must be relatively large. In particular, compared with Lemma 1, Lemma 3 indicates that, the 3PL must bear higher the traceability cost. This can be related to governmental subsidy in the FSCs. When the traceability cost is low, governmental subsidy will not make sense.

According to Lemma 3, the demand and optimal profits of the supplier, 3PL and retailer under the Model-3 are as follows:

$$D_3^{B*} = \frac{8bk}{64bk - \delta^2(14+b)},$$

$$\Pi_{S3}^* = \frac{128bk^2}{(64bk - \delta^2(14+b))^2},$$

$$\Pi_{L3}^* = \frac{k(8bk(512bk + \delta^2(12b + b^2 - 92)) + \delta^4(2-b)(14+b))}{2(8bk - \delta^2)(64bk - \delta^2(14+b))^2},$$

$$\Pi_{R3}^* = \frac{64bk^2}{(64bk - \delta^2(14+b))^2}.$$

Furthermore, we can derive the consumer surplus and social welfare are, respectively,

$$CS_3^* = \frac{32b^2k^2}{(64bk - \delta^2(14+b))^2},$$

$$SW_3^* = \frac{k(8bk(64bk(14+b) - \delta^2(36b + b^2 + 308)) + \delta^4(14+b)^2)}{2(8bk - \delta^2)(64bk - \delta^2(14+b))^2}.$$

### 4.4. Model-4

In this case, the government is not only adopting anti-pandemic strategy but also providing blockchain traceability subsidies in the post-pandemic era. The sequence of events is illustrated in Figure 1. The 3PL, supplier, and retailer's profits are, respectively,

$$\Pi_{L4} = m_4(1 - bp_4 + e_4 + \delta t_4) - \frac{1}{2}kt_4^2 + s_4 t_4, \tag{18}$$

$$\Pi_{S4} = (w_4 - m_4)(1 - bp_4 + e_4 + \delta t_4), \tag{19}$$

$$\Pi_{R4} = (p_4 - w_4)(1 - bp_4 + e_4 + \delta t_4). \tag{20}$$

In addition, the consumer surplus and social welfare can be formulated as, respectively,

$$CS_4 = \int_{bp_4 - e_4 - \delta t_4}^{1} (v - bp_4 + e_4 + \delta t_4)f(v)dv, \tag{21}$$

$$SW_4 = \Pi_{L4} + \Pi_{S4} + \Pi_{R4} + CS_4 - s_4 t_4 - \frac{1}{2}he_4^2. \tag{22}$$

Using the backward method to solve Equations (18)–(20) and (22) in turn, we can obtain the following lemma.

**Lemma 4.** *Under Model-4, if $k > \frac{(14+b)\delta^2}{64b}$ and $h > \frac{k(14+b)}{64bk - \delta^2(14+b)}$, the optimal strategies of the food supply chain are $s_4^* = \frac{hk\delta(6+b)}{64bkh - (h\delta^2+k)(14+b)}$, $e_4^* = \frac{k(14+b)}{64bkh - (h\delta^2+k)(14+b)}$, $w_4^* = \frac{48kh}{64bkh - (h\delta^2+k)(14+b)}$, $m_4^* = \frac{32kh}{64bkh - (h\delta^2+k)(14+b)}$, $p_4^* = \frac{56kh}{64bkh - (h\delta^2+k)(14+b)}$, $t_4^* = \frac{h\delta(14+b)}{64bkh - (h\delta^2+k)(14+b)}$, respectively.*

Lemma 4 shows that each member of the supply chain and the government have a unique optimal strategy only if $k > \frac{(14+b)\delta^2}{64b}$ and $h > \frac{k(14+b)}{64bk - \delta^2(14+b)}$. The condition $(k > \frac{(14+b)\delta^2}{64b})$ means that under Model-4, the cost of blockchain traceability must be relatively large and is the same as Model-3. The condition $(h > \frac{k(14+b)}{64bk - \delta^2(14+b)})$ means that under Model-4, the cost of the governmental anti-pandemic efforts must be relatively large. Compared with Lemma 2, Lemma 4 indicates that the government must bear higher the anti-pandemic cost. This can be also related to governmental subsidy in the blockchain-enabled FSCs.

By Lemma 4, the demand and optimal profits of the supplier, 3PL and retailer under the Model-4 are as follows:

$$D_4^{E*} = \frac{8bkh}{64bkh - (h\delta^2 + k)(14 + b)},$$

$$\Pi_{S4}^* = \frac{128bk^2h^2}{(64bkh - (h\delta^2 + k)(14 + b))^2},$$

$$\Pi_{L4}^* = \frac{kh^2(8bk(512bk + \delta^2(12b + b^2 - 92)) + \delta^4(2 - b)(14 + b))}{2(8bk - \delta^2)(64bkh - (h\delta^2 + k)(14 + b))^2},$$

$$\Pi_{R4}^* = \frac{64bk^2h^2}{(64bkh - (h\delta^2 + k)(14 + b))^2}.$$

Furthermore, we can derive the consumer surplus and social welfare are, respectively,

$$CS_4^* = \frac{32b^2k^2h^2}{(64bkh - (h\delta^2 + k)(14 + b))^2},$$

$$SW_4^* = \frac{kh(8bkh(64bk(14 + b) - \delta^2(36b + b^2 + 308)) - (14 + b)^2(8bk^2 - k\delta^2 - h\delta^4))}{2(8bk - \delta^2)(64bkh - (h\delta^2 + k)(14 + b))^2}.$$

By Lemmas 1–4, we can obtain the following corollaries.

**Corollary 1.** *(1)* $\frac{\partial s_3^*}{\partial \delta} > 0$, $\frac{\partial s_4^*}{\partial \delta} > 0$, $\frac{\partial e_2^*}{\partial \delta} > 0$, $\frac{\partial e_4^*}{\partial \delta} > 0$, $\frac{\partial D_j^*}{\partial \delta} > 0$, $\frac{\partial t_j^*}{\partial \delta} > 0$, and $\frac{\partial p_j^*}{\partial \delta} > \frac{\partial w_j^*}{\partial \delta} > \frac{\partial m_j^*}{\partial \delta} > 0$.
*(2)* $\frac{\partial \Pi_{Sj}^*}{\partial \delta} > 0$, $\frac{\partial \Pi_{Rj}^*}{\partial \delta} > 0$, $\frac{\partial CS_j^*}{\partial \delta} > 0$, $\frac{\partial \Pi_{Ln}^*}{\partial \delta} > 0$, and $\frac{\partial SW_n^*}{\partial \delta} > 0$, where $n = \{1, 2\}$.

Corollary 1-(1) shows that, when the 3PL adopts blockchain technology, the anti-pandemic effort level, subsidy level, demand, optimal prices and traceability level increase with the consumer preference for traceability information $\delta$. In particular, for the optimal prices, it has the greatest impact on wholesale prices and the least impact on logistics service prices. This is because, as the consumer preference for traceability information increases, supply chain members hope to obtain more product information to meet the market demand, thereby increasing the cost of 3PL. Thus, the 3PL would increase his prices to improve his profits. Then, the supplier and retailer would increase their prices to improve their profits. At the same time, the government will increase the subsidy level to encourage the 3PL to improve the traceability information level. This is similar to Wu et al. [1]. Interestingly, since the benefits of consumer preference for traceability information far outweigh the costs of increased anti-pandemic efforts, the governmental anti-pandemic effort level will also increase.

Corollary 1-(2) shows that, the profits of members, consumer surplus and social welfare increase with the consumer preference for traceability information. Therefore, the government should improve consumers' awareness of epidemic prevention, thereby increasing consumer preference for product information traceability, which can ultimately improve the profit of supply chain and the overall social benefits.

**Corollary 2.** *(1)* $\Pi_{Sj}^* = 2\Pi_{Rj}^*$.
*(2) If* $\frac{\delta^2}{8b} < k \leq \frac{\delta^2}{6b}$, *then* $\Pi_{Ln}^* \leq \Pi_{Rn}^* < \Pi_{Sn}^*$; *if* $\frac{\delta^2}{6b} < k \leq \frac{\delta^2}{4b}$, *then* $\Pi_{Rn}^* < \Pi_{Ln}^* \leq \Pi_{Sn}^*$; *if* $k > \frac{\delta^2}{4b}$, *then* $\Pi_{Rn}^* < \Pi_{Sn}^* < \Pi_{Ln}^*$, *where* $n = \{1, 2\}$.

Corollary 2 shows that, in any case, the profit of the supplier is always twice the profit of the retailer. This can be related to their rights and status in the multi-level supply chain. When governmental subsidy is not considered, the relationship between the profit of the 3PL and the profits of the supplier and retailer are affected by the traceability cost

coefficient $k$. Specifically, when the traceability cost coefficient is relatively large ($k > \frac{\delta^2}{4b}$), the profit of the 3PL is always higher than the profits of the supplier and retailer. To our common knowledge, the smaller $k$ leads to the smaller blockchain traceability cost and the higher profit for the 3PL. But, surprisingly, from Corollary 2, when there is no governmental subsidy, if $k$ is small, the profit of 3PL is smaller than that of the supplier and retailer; if $k$ is large, the profit of 3PL is higher. So, enterprises should adopt a much more proactive approach to implement blockchain traceability even when the cost coefficient is high and there is no governmental subsidy.

When considering governmental subsidy, the relationship between the profits of supply chain members is not clear, therefore, we will analyze their relationship through numerical analysis. As in Wu et al. [1] and Yang et al. [6], we use the following basic parameter values for our analysis: $b = 0.5$, $k = 5$, $h = 15$. Figure 2 shows that, the profit of the 3PL increases with the consumer preference for traceability information $\delta$. Moreover, when considering governmental subsidy strategy and regardless of whether the governmental anti-pandemic strategy is considered, the profit of the 3PL is always higher than the profits of the supplier and retailer. Also, from Corollary 2-(1), the profit of the retailer is the smallest (i.e., $\Pi_{L3}^* > \Pi_{S3}^* > \Pi_{R3}^*$ and $\Pi_{L4}^* > \Pi_{S4}^* > \Pi_{R4}^*$). The leader gains the most. Thus, this is consistent with the traditional wisdom.

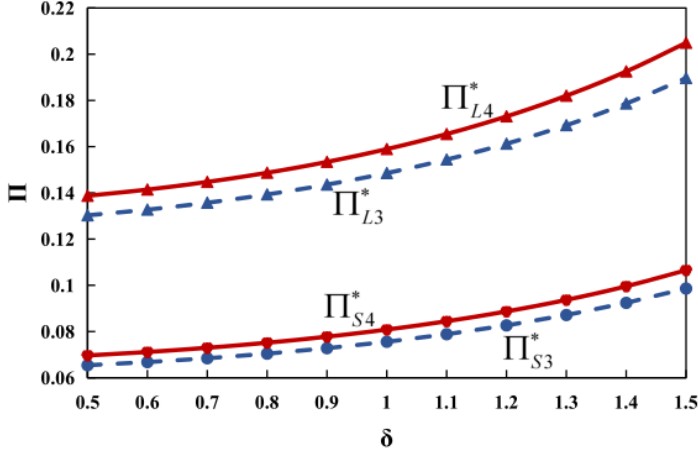

**Figure 2.** The impact of $\delta$ on the profits of supply chain members.

## 5. Model Comparison

In this section, we mainly explore three questions. First, under the scenario of with or without governmental subsidy, we explore whether the government should adopt anti-pandemic strategy in the post-pandemic era, and how the anti-pandemic strategy affects the optimal decisions and profits of supply chain members (i.e., Model-1 vs. Model-2, Model-3 vs. Model-4). Then, under the scenario of with or without governmental anti-pandemic strategy, we explore whether the government should provide subsidies for the 3PL with blockchain traceability in the post-pandemic era, and how the subsidy strategy affects optimal decisions and profits (i.e., Model-1 vs. Model-3, Model-2 vs. Model-4). Finally, when the government only adopts subsidy or anti-pandemic strategy, which strategy is more beneficial to supply chain members, consumers and society (i.e., Model-2 vs. Model-3).

### 5.1. Analysis of Governmental Anti-Pandemic Strategy

By comparing Model-1 and -2, Model-3 and -4, the following three propositions can be obtained.

**Proposition 1.** *(1) When governmental subsidy strategy is not considered, $t_2^* > t_1^*$, $w_2^* > w_1^*$, $m_2^* > m_1^*$, $p_2^* > p_1^*$, and $D_2^{E*} > D_1^{B*}$.*
*(2) When considering governmental subsidy strategy, $s_4^* > s_3^*$, $t_4^* > t_3^*$, $w_4^* > w_3^*$, $m_4^* > m_3^*$, $p_4^* > p_3^*$, and $D_4^{E*} > D_3^{B*}$.*

Proposition 1 shows that, regardless of whether the government provides subsidies for blockchain technology, the demand, optimal prices and traceability level under the anti-pandemic strategy are higher than those without the anti-pandemic strategy. When considering governmental subsidy strategy, the subsidy level under the anti-pandemic strategy is also higher. The reason is that, when the government implements the anti-pandemic strategy, it can improve the utility of consumers and thus increase market demand. In order to obtain greater benefits, the retailer will increase retail prices, which in turn will allow the supplier to increase wholesale prices and the 3PL to increase logistics service prices. At this time, the 3PL will have more funds and incentives to improve the traceability level to further increase demand. At the same time, the government will have more funds to provide subsidies and encourage the 3PL to improve the traceability level.

**Proposition 2.** *When governmental subsidy strategy is not considered, (1) $\Pi_{S2}^* > \Pi_{S1}^*$, $\Pi_{L2}^* > \Pi_{L1}^*$, $\Pi_{R2}^* > \Pi_{R1}^*$; (2) $CS_2^* > CS_1^*$, $SW_2^* > SW_1^*$.*

Proposition 2 shows that, when governmental subsidy strategy is not considered, the governmental anti-pandemic strategy can improve the profits of supply chain members, as well as consumer surplus and social welfare, so as to achieve a win-win situation for all parties. It can be seen from Proposition 1 that under the governmental anti-pandemic strategy, the market demand and prices will increase, which will increase the profits of the supplier and retailer. For the 3PL, the benefits brought by the increase in logistics service prices and traceability level are greater than the increase in traceability cost, so the profit of 3PL will be increased. An increase in consumers' utility results in an increase in consumer surplus. The benefits of the governmental anti-pandemic strategy to the supply chain are far greater than the cost of implementing the anti-pandemic strategy, thereby improving social welfare. Therefore, even in the post-pandemic era, the government should not give up anti-pandemic efforts, the benefits of which far outweigh the costs when governmental subsidy strategy is not considered.

**Proposition 3.** *When considering governmental subsidy strategy, (1) $\Pi_{S4}^* > \Pi_{S3}^*$, $\Pi_{L4}^* > \Pi_{L3}^*$, $\Pi_{R4}^* > \Pi_{R3}^*$; (2) $CS_4^* > CS_3^*$; (3) If $\frac{k(14+b)}{64bk-\delta^2(14+b)} < h \leq \frac{kA(14+b)}{64bk-\delta^2(14+b)}$, then $SW_4^* \leq SW_3^*$; if $h > \frac{kA(14+b)}{64bk-\delta^2(14+b)}$, then $SW_4^* > SW_3^*$, where $A = \frac{8bk(64bk(14+b)-\delta^2(b^2+36b+308))+\delta^4(14+b)^2}{(14+b)(8bk-\delta^2)(64bk-\delta^2(14+b))+128bk\delta^2}$.*

Proposition 3 shows that, when the government provides subsidy strategy for blockchain traceability, similar to Proposition 2, the governmental anti-pandemic strategy can improve the profits of supply chain members and consumer surplus. Interestingly, a lower cost coefficient of anti-pandemic effort $h$ harms the social welfare under the anti-pandemic strategy. This is contrary to our common sense. The reason is that, on the one hand, the low cost coefficient of anti-pandemic effort will increase the level of anti-pandemic effort excessively, resulting in a sharp increase in the cost of anti-pandemic effort and anti-pandemic effect; on the other hand, this will lead the government to reduce the subsidy level a little, at this time, the traceability level and cost will be lower, and the utility of traceability to consumers is reduced. But, the positive effect of the increase of anti-pandemic effort level is less than its negative effect. Ultimately, the social welfare is reduced. This shows that when considering the governmental subsidy strategy, the low cost of anti-pandemic effort will reduce social welfare, that is, the anti-pandemic strategy can be not good for the whole society. Comparing Propositions 2 and 3, it can be found that governmental subsidy may weaken the social benefits of government anti-pandemic strategy. Therefore, the government should combine anti-pandemic with subsidy strategies and keep both strategies at a reasonable level.

*5.2. Analysis of Governmental Subsidy Strategy*

By comparing Model-1 and -3, Model-2 and -4, the following three propositions can be obtained.

**Proposition 4.** *(1) When the governmental anti-pandemic strategy is not considered, $t_3^* > t_1^*$, $w_3^* > w_1^*$, $m_3^* > m_1^*$, $p_3^* > p_1^*$, and $D_3^{B*} > D_1^{B*}$.*
*(2) When considering the governmental anti-pandemic strategy, $e_4^* > e_2^*$, $t_4^* > t_2^*$, $w_4^* > w_2^*$, $m_4^* > m_2^*$, $p_4^* > p_2^*$, and $D_4^{E*} > D_2^{E*}$.*

Proposition 4 shows that, regardless of whether the government implements the anti-pandemic strategy, the demand, optimal prices and traceability level under the subsidy strategy are higher than those without the subsidy strategy. When considering the governmental anti-pandemic strategy, the level of anti-pandemic effort under the subsidy strategy is also higher. The reason is that, when the government implements subsidy strategy, it can incentivize the 3PL to improve the level of blockchain traceability, thus the market demand is increased. Similar to Proposition 1, in order to obtain greater benefits, the retailer will increase retail prices, which in turn will allow the supplier to increase wholesale prices. The 3PL will also increase the logistics service prices in order to ensure its own profits and recover the increased traceability cost. In order to prevent the excessively high subsidy level leading to a sharp increase in the cost of traceability and a decrease in demand, the government will increase the anti-pandemic effort level when considering the governmental subsidy strategy.

**Proposition 5.** *When the governmental anti-pandemic strategy is not considered, (1) $\Pi_{S3}^* > \Pi_{S1}^*$, $\Pi_{L3}^* > \Pi_{L1}^*$, $\Pi_{R3}^* > \Pi_{R1}^*$; (2) $CS_3^* > CS_1^*$, $SW_3^* > SW_1^*$.*

Proposition 5 shows that, when the governmental anti-pandemic strategy is not considered, the governmental subsidy strategy can improve the profits of supply chain members, as well as consumer surplus and social welfare, so as to achieve a win-win situation for all parties. This is similar to Proposition 2.

**Proposition 6.** *When considering the governmental anti-pandemic strategy, (1) $\Pi_{S4}^* > \Pi_{S2}^*$, $\Pi_{R4}^* > \Pi_{R2}^*$; (2) $CS_4^* > CS_2^*$; (3) If $h > \max\left\{ \frac{k(14+b)(bkB-(8bk-\delta^2)^2)-kB\delta^2}{(B+\delta^2(14+b)-64bk)(8bk-\delta^2)^2}, \frac{k(14+b)}{64bk-\delta^2(14+b)} \right\}$, then $\Pi_{L4}^* > \Pi_{L2}^*$, otherwise $\Pi_{L4}^* \leq \Pi_{L2}^*$, where $B = \sqrt{8bk(512bk + \delta^2(12b + b^2 - 92)) + \delta^4(2 - b)(14 + b)}$.*

Proposition 6 shows that, when the government adopts anti-pandemic strategy, similar to Proposition 5, the governmental subsidy strategy can improve the profits of the supplier and retailer, as well as consumer surplus. When the cost coefficient of anti-pandemic effort $h$ is large, the profit of 3PL under the subsidy strategy is higher. The reason is that, the high cost coefficient of anti-pandemic effort will increase the cost of anti-pandemic effort for the government. Therefore, in order to ensure high social welfare, the government will reduce the anti-pandemic effort level. At this time, the market demand will also decrease. However, when the government adopts the subsidy strategy, the government can encourage the 3PL to improve the traceability level through subsidy, so as to achieve an increase in demand. The increase in benefits brought about by the increase in subsidy level and prices is greater than the increase in traceability cost. Therefore, the profit of the 3PL becomes larger. Comparing Propositions 5 and 6, it can be found that the governmental anti-pandemic strategy may weaken the benefits of governmental subsidy to the 3PL.

As can be seen from Figure 3, the social welfare under the subsidy strategy is always higher than that without the subsidy strategy (i.e., $SW_4^* > SW_2^*$). The parameter settings are the same as above. Moreover, the social welfare increases with the consumer preference for traceability information $\delta$. Therefore, as long as the conditions for the increase of the 3PL's profit are met, the subsidy strategy can achieve a win-win situation for all parties and improve the overall social benefits when considering the governmental anti-pandemic strategy. Combining Proposition 6 and Figure 3, we find that the higher the governmental anti-pandemic cost coefficient, the higher the total benefit to the supply chain, and vice versa.

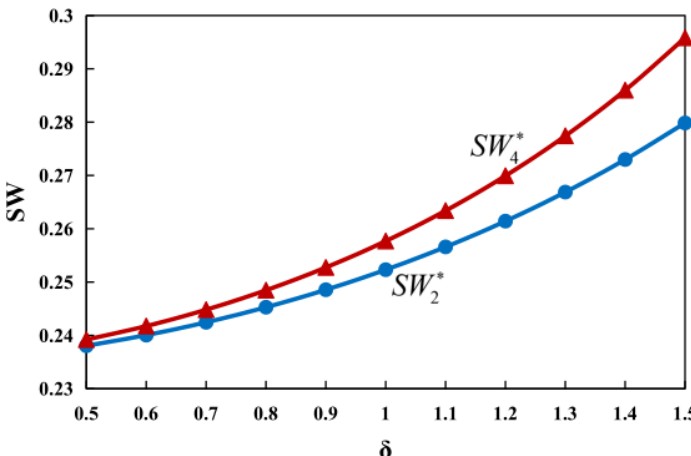

**Figure 3.** The impact of $\delta$ on the social welfare.

*5.3. Comparative Analysis of Governmental Anti-Pandemic and Subsidy Strategies*

In this subsection, we consider which strategy is more beneficial to supply chain members, consumers, and society when the government adopts only one strategy, i.e., compare the effects of the two strategies (anti-pandemic and subsidy strategies) in the post-pandemic era. By comparing Model-2 and -3, the following propositions are obtained.

**Proposition 7.** *(1) If $\frac{(14+b)\delta^2}{64b} < k \leq \frac{8}{14+b}$, then $t_3^* < t_2^*$.*
*(2) If $k > \max\left\{\frac{8}{14+b}, \frac{(14+b)\delta^2}{64b}\right\}$, then when $h > \max\left\{\frac{k(14+b)(14bk-\delta^2+b^2k)}{8b(k(14+b)-8)(8bk-\delta^2)}, \frac{k(14bk-\delta^2+b^2k)}{(8bk-\delta^2)^2}\right\}$, $t_3^* > t_2^*$, otherwise $t_3^* \leq t_2^*$.*

Proposition 7 shows that, the traceability information level based on blockchain technology is related to the cost coefficient of traceability and anti-pandemic effort (i.e., *k* and *h*). Specifically, when the cost coefficient of traceability *k* is small, the traceability level under the subsidy strategy is lower than that under the anti-pandemic strategy. Surprisingly, this shows that, compared with the governmental anti-pandemic strategy, the governmental subsidy for blockchain traceability will actually reduce the traceability level of the 3PL. The government should first decide which strategy to adopt based on the cost coefficient of traceability, otherwise it will affect the application of blockchain technology. When the cost coefficient of traceability is large, only when the cost coefficient of anti-pandemic effort *h* is greater than a threshold, the traceability level under the subsidy strategy is higher than that under the anti-pandemic strategy. The governmental anti-pandemic effort has increased market demand and increased retail prices. In order to obtain more profits, the 3PL will increase the traceability level. At the same time, the governmental subsidy can also promote the 3PL to improve the traceability level, but when the cost coefficient of traceability is small, the governmental subsidy is also lower, so that the traceability level does not increase much. Therefore, the traceability level is lower under the subsidy strategy. However, when the cost coefficient of traceability is large, the governmental subsidy is also higher, so that the traceability level increases. At the same time, due to the large cost coefficient of anti-pandemic effort and the lower level of anti-pandemic effort, the demand and retail price are not high. In order to reduce cost, the 3PL will reduce the traceability level. Therefore, the traceability level is higher under the subsidy strategy.

**Proposition 8.** *(1) If $\frac{k(14bk-\delta^2+b^2k)}{(8bk-\delta^2)^2} < h < \frac{8k(14bk-\delta^2+b^2k)}{\delta^2(6+b)(8bk-\delta^2)}$, then $w_3^* < w_2^*$, $m_3^* < m_2^*$, $p_3^* < p_2^*$, $D_3^{B*} < D_2^{E*}$, $\Pi_{S3}^* < \Pi_{S2}^*$, $\Pi_{R3}^* < \Pi_{R2}^*$, and $CS_3^* < CS_2^*$;*
*(2) If $h \geq \frac{8k(14bk-\delta^2+b^2k)}{\delta^2(6+b)(8bk-\delta^2)}$, then $w_3^* \geq w_2^*$, $m_3^* \geq m_2^*$, $p_3^* \geq p_2^*$, $D_3^{B*} \geq D_2^{E*}$, $\Pi_{S3}^* \geq \Pi_{S2}^*$, $\Pi_{R3}^* \geq \Pi_{R2}^*$, and $CS_3^* \geq CS_2^*$.*

Proposition 8 shows that, when the cost coefficient of anti-pandemic effort $h$ is small, the demand, optimal prices, profits of the supplier and retailer and consumer surplus under the subsidy strategy are lower than those under the anti-pandemic strategy. On the contrary, when the cost coefficient of anti-pandemic effort is greater than a threshold, the demand, optimal prices, profits of the supplier and retailer and consumer surplus under the subsidy strategy are higher than those under the anti-pandemic strategy. The smaller the cost coefficient of anti-pandemic effort, the lower the anti-pandemic cost. The government will invest in a higher level of anti-pandemic effort. At this time, the market demand and optimal prices will increase, thereby the profits of the supplier and retailer and consumer surplus will also be higher. The profit of 3PL and social welfare are directly related to the traceability level and subsidy level. Correlation comparisons are complex, therefore, we will next compare them through numerical analysis. It is illustrated in Figure 4, and the parameter settings are the same as above.

In Figure 4, Region I represents that $\Pi_{L2}^* > \Pi_{L3}^*$ and $SW_2^* > SW_3^*$, i.e., both the government and 3PL prefer to choose anti-pandemic strategy. Region II represents that $\Pi_{L2}^* < \Pi_{L3}^*$ and $SW_2^* > SW_3^*$, i.e., the government prefers to choose anti-pandemic strategy, but the 3PL prefers to choose subsidy strategy. Region III represents that $\Pi_{L2}^* < \Pi_{L3}^*$ and $SW_2^* < SW_3^*$, i.e., both the government and 3PL prefer to choose subsidy strategy. Figure 4 shows that, when the cost coefficient of anti-pandemic effort $h$ and the consumer preference for traceability information $\delta$ are low, the cost of anti-pandemic and the traceability effect of blockchain are low. At this time, the benefits brought by anti-pandemic strategy are higher than those brought by the subsidy strategy. Therefore, the government will choose the anti-pandemic strategy. The 3PL will also choose the anti-pandemic strategy to avoid high traceability cost. On the contrary, when the cost coefficient of anti-pandemic effort $h$ and the consumer preference for traceability information $\delta$ are high, the cost of anti-pandemic is high and the benefits brought by traceability are high. At this time, the subsidy strategy can motivate the 3PL to improve the traceability level. Ultimately, the subsidy strategy brings greater benefits than the anti-pandemic strategy. Since the government is the leader in the choice of anti-pandemic and subsidy strategies, in order to maximize social benefits, the government is very likely to adopt an anti-pandemic strategy instead of a subsidy strategy. Only when the consumer preference for traceability information is improved will the government adopt a subsidy strategy. Therefore, combining Proposition 8, we know that when the cost coefficient of anti-pandemic effort $h$ and the consumer preference for traceability information $\delta$ are low, the anti-pandemic strategy can make supply chain members, consumers and governments win-win. On the contrary, the subsidy strategy is more beneficial to all parties.

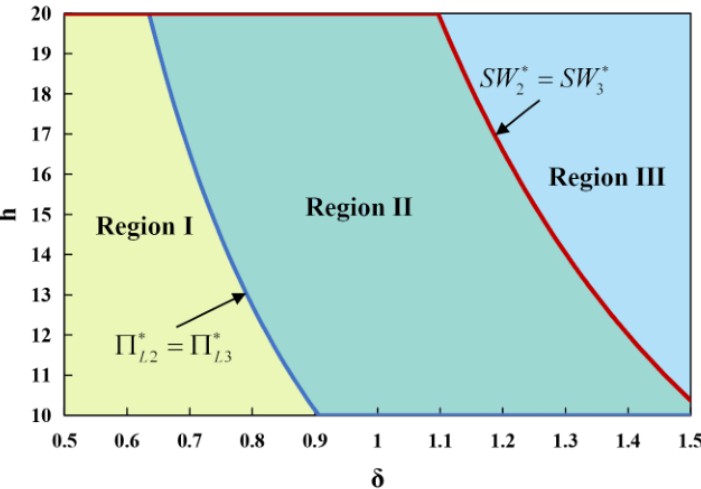

**Figure 4.** The choice of anti-pandemic and subsidy strategies for the government and 3PL.

## 6. Implications

Our paper has important theoretical and practical implications. Academically, in the context of considering the use of blockchain technology for product information traceability by the 3PL, this paper examines the value of governmental strategies for supply chain members, consumers and society, and explores whether the government chooses anti-pandemic or subsidy strategy in the post-pandemic era. To the best of our knowledge, this is the first attempt to study the impact of the governmental strategy selection on the pricing for food, the profits of the supply chain members, the consumer surplus and social welfare in the post-pandemic era. We quantitatively incorporate blockchain technology and consumer preference for traceability information into food traceability, which is our main contribution in modeling. In this way, we have done some preliminary studies on food supply chain traceability and epidemic prevention, which will help relevant researchers find follow-up research topics.

In practice, food safety is an important issue that needs attention from all parties, especially in the post-pandemic era. Governmental strategies and technological investment by enterprises are key to solving this problem. From our research results, we can obtain the following important managerial implications:

On the one hand, the government should raise consumers' awareness of epidemic prevention, thereby increasing consumer preference for product information traceability, and ultimately improving supply chain and overall social benefits. The governmental strategies are essential for recovery of supply chain that has experienced the impact of the epidemic, and even in the post-pandemic era, government should not abandon epidemic prevention measures. It is best to adopt anti-pandemic and subsidy strategies simultaneously. Only by promoting the joint efforts of government and enterprises to fight the epidemic is the best choice.

On the other hand, as the leader of the supply chain, the 3PL should improve the traceability information level based on blockchain technology to cooperate with the governmental strategies, especially when the consumer preference for traceability information is relatively high. When the cost coefficient of anti-pandemic effort is relatively small and the government has implemented an anti-pandemic strategy, then the 3PL should not accept governmental subsidy. Only by vigorously supporting the government to fight the epidemic can more profits be made. For other parties, such as the supplier, retailer and consumers, more profits can be obtained by actively cooperating and raising prices.

## 7. Conclusions

Our results can be summarized as:

First, in the four scenarios, the anti-pandemic effort level and subsidy level of the government, the demand, the optimal prices and traceability level increase with the consumer preference for traceability information, and the profits of members, consumer surplus and social welfare also increase with the consumer preference for traceability information. Moreover, the governmental subsidy strategy can make blockchain-enabled 3PL the most profitable and the retailer is the least profitable. Surprisingly, regardless of whether the government implements the anti-pandemic strategy, when governmental subsidy strategy is not considered, the smaller the cost coefficient of traceability, the profit of 3PL is not larger than that of the supplier and retailer, but smaller.

Second, regardless of whether the government provides subsidies for blockchain traceability, the demand, optimal prices, traceability level, profits of supply chain members and consumer surplus under the anti-pandemic strategy are higher than those without the anti-pandemic strategy. Specifically, when the governmental subsidy strategy is not considered, the governmental anti-pandemic strategy can improve social welfare, so as to achieve a win-win situation. When the government provides subsidies for traceability, the subsidy level under the anti-pandemic strategy is higher, but only when the cost coefficient of anti-pandemic effort is large, the social welfare is higher, otherwise the social welfare

under the anti-pandemic strategy is lower. Moreover, the governmental subsidy may weaken the social benefits of governmental anti-pandemic strategy.

Third, regardless of whether the government implements the anti-pandemic strategy, the demand, optimal prices, traceability level, profits of the supplier and retailer, consumer surplus and social welfare under the subsidy strategy are higher than those without the subsidy strategy. Moreover, when the governmental anti-pandemic strategy is not considered, the governmental subsidy strategy can improve the profit of 3PL. However, considering the governmental anti-pandemic strategy, only when the cost coefficient of anti-pandemic effort is large, the profit of 3PL under the subsidy strategy is higher. The government is willing to pay a greater anti-pandemic effort level under the subsidy strategy. In addition, we find that when the governmental anti-pandemic strategy is considered, the higher the governmental anti-pandemic cost coefficient, the higher the total benefit to the supply chain, and vice versa. The governmental anti-pandemic strategy may weaken the benefits of governmental subsidy to the 3PL.

Finally, in the scenario where the government can only adopt one strategy, we find that when the cost coefficient of anti-pandemic effort is small, the demand, optimal prices, profits of the supplier and retailer and consumer surplus under the subsidy strategy are lower than those under the anti-pandemic strategy. The traceability level based on blockchain technology is related to the cost coefficient of traceability and anti-pandemic effort. Interestingly, when the cost coefficient of traceability is small, the governmental subsidy for blockchain traceability will actually reduce the traceability level of the 3PL. It shows that the government should first decide which strategy to adopt based on the cost coefficient of traceability, otherwise it will affect the application of blockchain technology. When the cost coefficient of anti-pandemic effort and the consumer preference for traceability information are low, the anti-pandemic strategy can make all parties win-win. On the contrary, the subsidy strategy is more beneficial to all parties.

This study has several limitations and possible extensions. First, we only consider that the 3PL bears the blockchain traceability cost and determines the traceability level, but Yang et al. [6] showed that when the supply chain members participate in the application and decision-making of the blockchain, the supply chain profit is the largest. Therefore, the next thing to consider is a scenario where both the supplier and retailer adopt blockchain traceability. Second, we only consider governmental subsidy for the 3PL using blockchain traceability. In the process of epidemic prevention, the supplier and retailer also play a large role. Considering subsidies for all members of the supply chain is also worth exploring [7]. We hope that this paper can inspire more research work on the application of blockchain technology and governmental epidemic prevention.

**Author Contributions:** Formal analysis, C.L.; methodology, Q.L.; supervision, Y.S.; writing—original draft, C.L.; writing—review and editing, Q.L. All authors have read and agreed to the published version of the manuscript.

**Funding:** This research is supported by the Zhejiang Provincial Natural Sciences Foundation of China (Nos. LY22G020007) and Natural Science Foundation of China (Nos. 71672179).

**Institutional Review Board Statement:** Not applicable.

**Informed Consent Statement:** Informed consent was obtained from all subjects involved in the study.

**Data Availability Statement:** Not applicable.

**Conflicts of Interest:** The authors declare no conflict of interest.

## Nomenclature

| Notations | Descriptions |
| --- | --- |
| Decision variables | |
| $p$ | The retail price per unit product |
| $w$ | The wholesale price per unit product |
| $m$ | The 3PL's logistics service price per unit product |
| $t$ | The traceability information level based on blockchain technology, hereafter traceability level |
| $e$ | The governmental anti-pandemic effort level |
| $s$ | The governmental subsidy level per unit traceability level |
| Parameters | |
| $b$ | The price sensitivity of the product, $b > 0$ |
| $k$ | The cost coefficient of traceability by using blockchain technology, $k > 0$ |
| $h$ | The cost coefficient of anti-pandemic effort, $h > 0$ |
| $\delta$ | The consumer preference for traceability information, $\delta > 0$ |
| $D$ | The market demand |
| $\Pi_L$ | The 3PL's profit function |
| $\Pi_S$ | The supplier's profit function |
| $\Pi_R$ | The retailer's profit function |
| $CS$ | The consumer surplus |
| $SW$ | The social welfare |

## Appendix A

**Proof to Lemma 1.** By Equation (5), we can determine first-order and second-order derivatives of $\Pi_{R1}$ with respect to $p_1$, i.e., $\frac{\partial \Pi_{R1}}{\partial p_1} = 1 - 2bp_1 + \delta t_1 + bw_1$ and $\frac{\partial^2 \Pi_{R1}}{\partial p_1^2} = -2b$. Obviously, $\frac{\partial^2 \Pi_{R1}}{\partial p_1^2} < 0$ holds. By first-order condition $\frac{\partial \Pi_{R1}}{\partial p_1} = 0$, we have

$$p_1 = \frac{1 + \delta t_1 + bw_1}{2b}. \tag{A1}$$

Substituting Equation (A1) into Equation (4), we can determine first-order and second-order derivatives of $\Pi_{S1}$ with respect to $w_1$, i.e., $\frac{\partial \Pi_{S1}}{\partial w_1} = \frac{1 - 2bw_1 + \delta t_1 + bm_1}{2}$ and $\frac{\partial^2 \Pi_{S1}}{\partial w_1^2} = -b$. Obviously, $\frac{\partial^2 \Pi_{S1}}{\partial w_1^2} < 0$ holds. By first-order condition $\frac{\partial \Pi_{S1}}{\partial w_1} = 0$, we have

$$w_1 = \frac{1 + \delta t_1 + bm_1}{2b}. \tag{A2}$$

Substituting Equations (A1) and (A2) into Equation (3), we can determine first-order and second-order derivatives of $\Pi_{L1}$ with respect to $m_1$ and $t_1$, i.e., $\frac{\partial \Pi_{L1}}{\partial m_1} = \frac{1 + \delta t_1 - 2bm_1}{4}$, $\frac{\partial^2 \Pi_{L1}}{\partial m_1^2} = -\frac{b}{2}$, $\frac{\partial \Pi_{L1}}{\partial t_1} = \frac{m_1 \delta - 4kt_1}{4}$, $\frac{\partial^2 \Pi_{L1}}{\partial t_1^2} = -k$, $\frac{\partial^2 \Pi_{L1}}{\partial m_1 \partial t_1} = \frac{\partial^2 \Pi_{L1}}{\partial t_1 \partial m_1} = \frac{\delta}{4}$, then the Hessian matrix is

$$H(m_1, t_1) = \begin{bmatrix} \frac{\partial^2 \Pi_{L1}}{\partial m_1^2} & \frac{\partial^2 \Pi_{L1}}{\partial m_1 \partial t_1} \\ \frac{\partial^2 \Pi_{L1}}{\partial t_1 \partial m_1} & \frac{\partial^2 \Pi_{L1}}{\partial t_1^2} \end{bmatrix} = \begin{bmatrix} -\frac{b}{2} & \frac{\delta}{4} \\ \frac{\delta}{4} & -k \end{bmatrix}. \tag{A3}$$

Obviously, $|H(m_1, t_1)| = \frac{8bk - \delta^2}{16}$, $\frac{\partial^2 \Pi_{L1}}{\partial m_1^2} < 0$ and $\frac{\partial^2 \Pi_{L1}}{\partial t_1^2} < 0$, only when $|H(m_1, t_1)| > 0$ is satisfied, i.e., $k > \frac{\delta^2}{8b}$, $H(m_1, t_1)$ is a negative definite matrix, which implies that $\Pi_{L1}$

jointly concave in $(m_1, t_1)$. The unique optimal $m_1^*$ and $t_1^*$ should satisfy the first-order conditions. Hence, we have

$$m_1^* = \frac{4k}{8bk - \delta^2}, t_1^* = \frac{\delta}{8bk - \delta^2}. \tag{A4}$$

Substituting Equation (A4) into Equations (A1) and (A2), we can determine the optimal strategies, i.e., $w_1^* = \frac{6k}{8bk - \delta^2}$, $p_1^* = \frac{7k}{8bk - \delta^2}$. Therefore, Lemma 1 holds. $\square$

**Proof to Lemma 2–4.** The proofs are similar to the proof of Lemma 1, so they are omitted. $\square$

**Proof to Corollary 1.** According to Lemma 1 and $D_1^{B*}$, we can determine first-order derivatives of the demand, the optimal prices and traceability level of the 3PL with respect to $\delta$, i.e.,

$$\frac{\partial D_1^{B*}}{\partial \delta} = \frac{2\delta bk}{(8bk - \delta^2)^2} > 0, \tag{A5}$$

$$\frac{\partial t_1^*}{\partial \delta} = \frac{8bk + \delta^2}{(8bk - \delta^2)^2} > 0, \tag{A6}$$

$$\frac{\partial p_1^*}{\partial \delta} = \frac{14\delta k}{(8bk - \delta^2)^2} > 0, \tag{A7}$$

$$\frac{\partial m_1^*}{\partial \delta} = \frac{8\delta k}{(8bk - \delta^2)^2} > 0, \tag{A8}$$

$$\frac{\partial w_1^*}{\partial \delta} = \frac{12\delta k}{(8bk - \delta^2)^2} > 0. \tag{A9}$$

Likewise, we can obtain the similar results from Lemmas 2–4, the demand, supply members' profits, consumer surplus and social welfare. Hence, (1) and (2) of Corollary 1 hold. $\square$

**Proof to Corollary 2.** From $\Pi_{S1}^*$ and $\Pi_{R1}^*$, obviously, $\Pi_{S1}^* = 2\Pi_{R1}^*$. Then, according to the profits of the supply chain members under the Model-1, we have

$$\Pi_{L1}^* - \Pi_{R1}^* = \frac{k(6bk - \delta^2)}{2(8bk - \delta^2)^2}, \tag{A10}$$

$$\Pi_{L1}^* - \Pi_{S1}^* = \frac{k(4bk - \delta^2)}{2(8bk - \delta^2)^2}. \tag{A11}$$

By Equation (A10), we can know that when $k > \frac{\delta^2}{6b}$, we have $\Pi_{L1}^* > \Pi_{R1}^*$, otherwise $\Pi_{L1}^* \leq \Pi_{R1}^*$. By Equation (A11), we can know that when $k > \frac{\delta^2}{4b}$, we have $\Pi_{L1}^* > \Pi_{S1}^*$, otherwise $\Pi_{L1}^* \leq \Pi_{S1}^*$. Finally, according to $\Pi_{S1}^* > \Pi_{R1}^*$, we have that if $\frac{\delta^2}{8b} < k \leq \frac{\delta^2}{6b}$, then $\Pi_{L1}^* \leq \Pi_{R1}^* < \Pi_{S1}^*$; if $\frac{\delta^2}{6b} < k \leq \frac{\delta^2}{4b}$, then $\Pi_{R1}^* < \Pi_{L1}^* \leq \Pi_{S1}^*$; if $k > \frac{\delta^2}{4b}$, then $\Pi_{R1}^* < \Pi_{S1}^* < \Pi_{L1}^*$. Likewise, we can obtain the similar results from the Model-2, Model-3 and Model-4. Hence, (1) and (2) of Corollary 2 hold. $\square$

**Proof to Proposition 1.** According to Lemmas 1 and 2, by comparing the traceability levels, optimal prices, and demand, respectively, we have

$$t_2^* - t_1^* = \frac{\delta k(14bk - \delta^2 + b^2k)}{(8bk - \delta^2)(h(8bk - \delta^2)^2 - k(14bk - \delta^2 + b^2k))},$$

$$w_2^* - w_1^* = \frac{6k^2(14bk - \delta^2 + b^2k)}{(8bk - \delta^2)(h(8bk - \delta^2)^2 - k(14bk - \delta^2 + b^2k))},$$

$$m_2^* - m_1^* = \frac{4k^2(14bk - \delta^2 + b^2k)}{(8bk - \delta^2)(h(8bk - \delta^2)^2 - k(14bk - \delta^2 + b^2k))},$$

$$p_2^* - p_1^* = \frac{7k^2(14bk - \delta^2 + b^2k)}{(8bk - \delta^2)(h(8bk - \delta^2)^2 - k(14bk - \delta^2 + b^2k))},$$

$$D_2^{E*} - D_1^{B*} = \frac{bk^2(14bk - \delta^2 + b^2k)}{(8bk - \delta^2)(h(8bk - \delta^2)^2 - k(14bk - \delta^2 + b^2k))}.$$

According to the conditions $k > \frac{\delta^2}{8b}$ and $h > \frac{k(14bk - \delta^2 + b^2k)}{(8bk - \delta^2)^2}$, we can obtain that $t_2^* - t_1^* > 0$, $w_2^* - w_1^* > 0$, $m_2^* - m_1^* > 0$, $p_2^* - p_1^* > 0$, $D_2^{E*} - D_1^{B*} > 0$. Thus, Corollary 2-(1) holds. Likewise, we can obtain the similar results from Lemmas 3 and 4, hence, Corollary 2-(2) also holds. □

**Proof to Proposition 2.** According to the supply members' profits, consumer surplus and social welfare, by comparing them, respectively, we have

$$\Pi_{R2}^* - \Pi_{R1}^* = \frac{bk^3(2h(8bk - \delta^2)^2 - k(14bk - \delta^2 + b^2k))(14bk - \delta^2 + b^2k)}{(8bk - \delta^2)^2(h(8bk - \delta^2)^2 - k(14bk - \delta^2 + b^2k))^2},$$

$$\Pi_{S2}^* - \Pi_{S1}^* = \frac{2bk^3(2h(8bk - \delta^2)^2 - k(14bk - \delta^2 + b^2k))(14bk - \delta^2 + b^2k)}{(8bk - \delta^2)^2(h(8bk - \delta^2)^2 - k(14bk - \delta^2 + b^2k))^2},$$

$$\Pi_{L2}^* - \Pi_{L1}^* = \frac{k^2(2h(8bk - \delta^2)^2 - k(14bk - \delta^2 + b^2k))(14bk - \delta^2 + b^2k)}{2(8bk - \delta^2)(h(8bk - \delta^2)^2 - k(14bk - \delta^2 + b^2k))^2},$$

$$CS_2^* - CS_1^* = \frac{b^2k^3(2h(8bk - \delta^2)^2 - k(14bk - \delta^2 + b^2k))(14bk - \delta^2 + b^2k)}{2(8bk - \delta^2)^2(h(8bk - \delta^2)^2 - k(14bk - \delta^2 + b^2k))^2},$$

$$SW_2^* - SW_1^* = \frac{k^2(14bk - \delta^2 + b^2k)^2}{2(8bk - \delta^2)^2(h(8bk - \delta^2)^2 - k(14bk - \delta^2 + b^2k))}.$$

According to the conditions $k > \frac{\delta^2}{8b}$ and $h > \frac{k(14bk - \delta^2 + b^2k)}{(8bk - \delta^2)^2}$, we can obtain that $\Pi_{R2}^* - \Pi_{R1}^* > 0$, $\Pi_{S2}^* - \Pi_{S1}^* > 0$, $\Pi_{L2}^* - \Pi_{L1}^* > 0$, $CS_2^* - CS_1^* > 0$, $SW_2^* - SW_1^* > 0$. Thus, Proposition 2 holds. □

**Proof to Proposition 3–8.** The proofs are similar to the proof of Propositions 1 and 2, so they are omitted. □

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
