# Peer review of "Governmental Anti-Pandemic and Subsidy Strategies for Blockchain-Enabled Food Supply Chains in the Post-Pandemic Era"

_sustainability, doi:10.3390/su14159497_

Round 1

Reviewer 1 Report

The research with a very interesting approach and high research quality, including a decision-making process, government interests , post -epidemic era (new term) and background for blockchain applications.

1-Please show the novelty of this research, the result and findings of this research need require the discussion using relevant theories and previous research.

2- Please show what is the contribution of the research to the parties who need these results in accordance with the objectives of this research.

3- Please show the " Practical and Theoretical implications" in a separate section before the "Conclusion" section. 

Author Response

Response to Reviewer 1#

Dear Reviewer,

We thank you very much and the review team for your thoughtful comments and valuable suggestions on our paper.  We have put in much efforts to address all your and reviewer's concerns in this revision. To address the review team’s concerns, we have modified the paper. The improvement of this paper will be impossible without your constructive suggestions.

In the revised manuscript, the major changed parts are shown in red fonts, and other small changed parts have also been revised. In the following, we first provide a summary of the main changes in this revision.

Revision Summary

  1. Reviews pointed out that show the novelty of this research. Therefore, we added research gap to highlight our theoretical contributions.
  2. Reviews pointed out that show what is the contribution of the research to the parties and show the “Practical and Theoretical implications” in a separate section. Therefore, we summarized the practical and theoretical implications of our article and wrote them in Section 6.
  3. Reviews pointed out that check grammatical bugs and typos, so we carefully check the whole manuscript and try to avoid any grammar or syntax error.
  4. Reviews pointed out that need to revise the referencing system, so we have carefully revised the reference system in the order in which the references are cited, and have reviewed and improved the citation and reference format of the literature.

Below are our point-to-point responses to your comments. Your original comments are highlighted in italics and our responses are in deep blue color.

  1. Please show the novelty of this research, the result and findings of this research need require the discussion using relevant theories and previous research.

[Response]

Thank you for your comments. In the revised manuscript, at the end of the literature review in Section 2, we add a subsection “2.4 Research gap” in Page 5. Specifically, we add the following paragraph.

" In this subsection, we summarize the research gaps between our work and the existing literature and further highlight our contributions. First, to the best of our knowledge, this paper is the first to use the game theory model to describe the importance of blockchain technology to food traceability in the context of the COVID-19. We explore the government's choice of two strategies for anti-epidemic and subsidy under the blockchain traceability. Second, this paper considers the 3PL as the leader of the supply chain, which bears the cost of blockchain traceability and determines the traceability level, and explores the impact of different government strategies on the traceability levels and profits. Third, considering the government's decision to maximize social welfare, we study the impact of different scenarios on society and consumers, so as to provide decision-making references for the government in the post-epidemic era and a new background for blockchain applications. This is not only conducive to the stability and sustainable development of the supply chain, but also to the mitigation of the epidemic."

Furthermore, in the last sentence of each subsection in the literature review section, we analyze the similarities and differences between our paper and the previous research. We marked them in red. Specifically, as follows:

In Subsection 2.1, we have a paragraph: “In view of the food supply chains with health-safety concerns in the post-pandemic era, considering the traceability and transparency of reliable information supported by blockchain implementation, we explore the impact of blockchain on operational decisions such as FSC pricing and traceability level, which enriches the existing research.”

In Subsection 2.2, we have a paragraph: “Similar to Yang et al. [6] and Wu et al. [1], we study a blockchain-based food traceability system dominated by the 3PL, taking into account consumers' concerns about epidemic infections and their preferences for traceability information. The difference with our research is that we focus on the government's strategic choices in the post-pandemic era..”

In Subsection 2.3, we have a paragraph: “We also consider the government subsidy strategy, but it is a subsidy for the 3PL using blockchain traceability. At the same time, we consider the government to decide the anti-epidemic effort level, and explore what strategies the government should adopt to deal with the epidemic in the post-epidemic era.”

  1. Please show what is the contribution of the research to the parties who need these results in accordance with the objectives of this research.

[Response]

Thank you for your comments. In the revised manuscript, at the end of the implications in Section 6, we summarize the practical value of our article to the government and supply chain members in Page 17. We marked them in red.

Specifically, as follows: " In practice, food safety is an important issue that needs attention from all parties, especially in the post-epidemic era. Government strategies and technological investment by enterprises are key to solving this problem. From our research results, we can obtain the following important managerial implications:

On the one hand, the government should raise consumers' awareness of epidemic prevention, thereby increasing consumers' preference for product information traceability, and ultimately improving supply chain and overall social benefits. The government strategy is essential for recovery of supply chain that has experienced the impact of the epidemic, and even in the post-epidemic era, government should not abandon epidemic prevention measures. It is best to adopt anti-epidemic and subsidy strategies simultaneously. Only by promoting the joint efforts of government and enterprises to fight the epidemic is the best choice.

On the other hand, as the leader of the supply chain, the 3PL should improve the traceability information level based on blockchain technology to cooperate with the government's strategies, especially when the consumers' preference for traceability information is relatively high. When the cost coefficient of anti-epidemic effort is relatively small and the government has implemented an anti-epidemic strategy, then the 3PL should not accept government subsidy. Only by vigorously supporting the government to fight the epidemic can more profits be made. For other parties, such as the supplier, retailer and consumers, more profits can be obtained by actively cooperating and raising prices. "

  1. Please show the " Practical and Theoretical implications" in a separate section before the "Conclusion" section.

[Response]

Thank you for your comments. In Page 17, we have shown the " Practical and Theoretical implications" in a separate section before the "Conclusion" section and denoted this as Section 6. We marked them in red.

Specifically, as follows: " Our paper has important theoretical and practical implications. Academically, in the context of considering the use of blockchain technology for product information traceability by the 3PL, this paper examines the value of government strategies for supply chain members, consumers and society, and explores whether the government chooses anti-epidemic or subsidy strategy in the post-epidemic era. To the best of our knowledge, this is the first attempt to study the impact of the government strategy selection on the pricing for food in the post-pandemic era, the profits of the supply chain members, the consumer surplus and social welfare. We quantitatively incorporate blockchain technology and consumers’ preference for traceability information into food traceability, which is our main contribution in modeling. In this way, we have done some preliminary studies on food supply chain traceability and epidemic prevention, which will help relevant researchers find follow-up research topics.

In practice, food safety is an important issue that needs attention from all parties, especially in the post-epidemic era. Government strategies and technological investment by enterprises are key to solving this problem. From our research results, we can obtain the following important managerial implications:

On the one hand, the government should raise consumers' awareness of epidemic prevention, thereby increasing consumers' preference for product information traceability, and ultimately improving supply chain and overall social benefits. The government strategy is essential for recovery of supply chain that has experienced the impact of the epidemic, and even in the post-epidemic era, government should not abandon epidemic prevention measures. It is best to adopt anti-epidemic and subsidy strategies simultaneously. Only by promoting the joint efforts of government and enterprises to fight the epidemic is the best choice.

On the other hand, as the leader of the supply chain, the 3PL should improve the traceability information level based on blockchain technology to cooperate with the government's strategies, especially when the consumers' preference for traceability information is relatively high. When the cost coefficient of anti-epidemic effort is relatively small and the government has implemented an anti-epidemic strategy, then the 3PL should not accept government subsidy. Only by vigorously supporting the government to fight the epidemic can more profits be made. For other parties, such as the supplier, retailer and consumers, more profits can be obtained by actively cooperating and raising prices. "

At last, we thank you very much for your appreciation of our paper. As you can see from the revision summary, our responses, and the revised paper, we have tried our best to justify our model, illustrate the research gap, avoid any grammar or syntax error, and improve our paper. We thank you so much again for your constructive suggestions, which have greatly improved the quality of this paper. We hope you will find this version to your satisfaction. Thanks!

Reviewer 2 Report

Congratulations - a very timely paper... However, as with any paper that deals with so many competing models, it can become tedious - but much detail is necessary. There is an element of repetition in several cases but these can be easily removed [see detailed comments below]. More important though is the need to revise the referencing system - not alphabetically but in the order in which references are cited. This will take some time...

L3     logistics provider

L4     We then use...

L5     ... subsidy level for the supply chain

L8     delete this

L9     anti-epidemic [what]????

L11   delete

keywords: choose alternative keywords - these are already in the title

Intro  first line - either rewrite or delete - its factually incorrect

L17   Due to the high...

L23   delete [and cold chain logistics]

L25   were infected

L27   delete

L28   rebuild consumer trust [and get out of the woods][delete] post the....

L40   logistics providers [delete chain]

         Alibaba has subsequently launched...

L47   ... the world, with food supply chains being deeply affected

         was severe...

L48   delete [have] delete [obvious]

L49   .. by the outbreak too [have taken][delete]

L50   delete have

L54   delete [and everything resumed as before][not true]

L56   ... more concerned about food safety..

L61   subsidized the...

L64   rewrite - blockchain technology does NOT prevent epidemics

L65   Blockchain traceability....

L68   ... provides subsidies to allow... delete [level]

L70   delete [these issues]

L74   In the past, 

L76-89  DELETE. These are results - they have NO place in the Introduction

L95   Our research...

L104  Section 5 provides a comparison... the different....

L105  Delete

L113  .... has been studied by many scholars

L121  ... highlighting where the use of ... made...

L123-125   Delete

L127   ...could increase...

L128   depended

L143   consumers are paying more attention to food safety which has aroused....

L150  in building a

L156  new paragraph

L159  ...on fresh food traceability and the impact of food marketing

L176-177  ????

L181  ..of policy implementation

L201-212   either cut and paste into the Intro or delete 

L345  ... level. This is...

L347  new paragraph

L350  ... of antiepidemic [what] - is it strategy or effort????

L356   any case

L357  ... of the retailer. This can be ....

L359  are affected

L429  antiepidemic [what]????

L448  , in order

L467  ... are higher

L504  delete much

L532  antiepidemic [what]??

L539  government is the leader????? How is this so????

L548-558. Delete and replace with  Our results can be summarized as:   

L582  ...scenario where the government

L592  antiepidemic [what]??

L598  ditto

L604  ditto

Author Response

Response to Reviewer 2#

Dear Reviewer,

We thank you very much and the review team for your thoughtful comments and valuable suggestions on our paper.  We have put in much efforts to address all your and reviewer's concerns in this revision. To address the review team’s concerns, we have modified the paper. The improvement of this paper will be impossible without your constructive suggestions.

In the revised manuscript, the major changed parts are shown in red fonts, and other small changed parts have also been revised. In the following, we first provide a summary of the main changes in this revision.

Revision Summary

  1. Reviews pointed out that show the novelty of this research. Therefore, we added research gap to highlight our theoretical contributions.
  2. Reviews pointed out that show what is the contribution of the research to the parties and show the “Practical and Theoretical implications” in a separate section. Therefore, we summarized the practical and theoretical implications of our article and wrote them in Section 6.
  3. Reviews pointed out that check grammatical bugs and typos, so we carefully check the whole manuscript and try to avoid any grammar or syntax error.
  4. Reviews pointed out that need to revise the referencing system, so we have carefully revised the reference system in the order in which the references are cited, and have reviewed and improved the citation and reference format of the literature.

Below are our point-to-point responses to your comments. Your original comments are highlighted in italics and our responses are in deep blue color.

  1. There is an element of repetition in several cases but these can be easily removed [see detailed comments below].

[Response]

Thanks for your great suggestion. According to your detailed comments below, we have revised the whole manuscript carefully and tried to avoid any grammar or syntax error. We hope that the language is now acceptable.

Specifically, the major changed parts are shown in red fonts, and other problematic writing have also been revised, but not all of them are marked in red. In addition, the two unmodified parts are explained as follows:

L76-89 DELETE. These are results - they have NO place in the Introduction”:  Referring to the writing method of the previous literature (“total-point-total” structure), here is a brief description of the tasks of this paper, and a brief summary of the key results to answer the questions raised above, giving readers a clear idea. Specifically in the text: "In order to solve the above problems, we first study four scenarios in which the government adopts an anti-epidemic strategy or a subsidy strategy. We find that the anti-epidemic strategy is only established when the cost coefficient of anti-epidemic effort is too large, and the subsidy strategy is established when the cost coefficient of traceability is too large. Then, we analyze the impact of the consumers' preference for traceability information on equilibrium solutions, the profits of supply chain member, consumer surplus and social welfare. The results show that, no matter in which scenario, the greater the consumers' preference for traceability information, the more beneficial to all parties and society. Finally, by comparing various scenarios, we find that the simultaneous adoption of anti-epidemic and subsidy strategies by the government is always beneficial to the supplier, retailer and consumers, and the impact on the third-party logistics provider (3PL) and social welfare is related to the cost coefficient of anti-epidemic effort and the consumers' preference for traceability information. In a word, it is most detrimental to supply chain members and society if the government does not adopt any strategy."

L201-212   either cut and paste into the Intro or delete”: Referring to the writing method of the previous literature, here is a summary of the differences between this paper and previous literature, further highlighting our theoretical contributions. In the revised manuscript, at the end of the literature review in Section 2, we added it on page 5 as the fourth subsection "Research gap". Specifically in the text: " In this subsection, we summarize the research gaps between our work and the existing literature and further highlight our contributions. First, to the best of our knowledge, this paper is the first to use the game theory model to describe the importance of blockchain technology to food traceability in the context of the COVID-19. We explore the government's choice of two strategies for anti-epidemic and subsidy under the blockchain traceability. Second, this paper considers the 3PL as the leader of the supply chain, which bears the cost of blockchain traceability and determines the traceability level, and explores the impact of different government strategies on the traceability levels and profits. Third, considering the government's decision to maximize social welfare, we study the impact of different scenarios on society and consumers, so as to provide decision-making references for the government in the post-epidemic era and a new background for blockchain applications. This is not only conducive to the stability and sustainable development of the supply chain, but also to the mitigation of the epidemic. "

  1. More important though is the need to revise the referencing system - not alphabetically but in the order in which references are cited.

[Response]

Thank you for your comments. In the revised manuscript, we have carefully revised the reference system in the order in which the references are cited, and have reviewed and improved the citation and reference format of the literature. We hope that the reference format is now acceptable.

At last, we thank you very much for your appreciation of our paper. As you can see from the revision summary, our responses, and the revised paper, we have tried our best to justify our model, illustrate the research gap, avoid any grammar or syntax error, and improve our paper. We thank you so much again for your constructive suggestions, which have greatly improved the quality of this paper. We hope you will find this version to your satisfaction. Thanks!